# Using decoys and camera traps to estimate depredation rates and neonate survival

Hailey M. Boone[1¤]*, Krishna Pacifici[1], Christopher E. Moorman[1], Roland Kays[1,2]

1 Fisheries, Wildlife, and Conservation Biology Program, North Carolina State University, Raleigh, NC, United States of America, 2 North Carolina Museum of Natural Sciences, Raleigh, NC, United States of America

ⓧ These authors contributed equally to this work.
¤ Current address: Department of Fisheries and Wildlife, Michigan State University, East Lansing, MI, United States of America
* boonehai@msu.edu

**Data Availability Statement:** All relevant data are within the paper and its Supporting Information files.

**Funding:** The author(s) received no specific funding for this work.

## Abstract

Ungulate neonates—individuals less than four weeks old—typically experience the greatest predation rates, and variation in their survival can influence ungulate population dynamics. Typical methods to measure neonate survival involve capture and radio-tracking of adults and neonates to discover mortality events. This type of fieldwork is invasive and expensive, can bias results if it leads to neonate abandonment, and may still have high uncertainty about the predator species involved. Here we explore the potential for a non-invasive approach to estimate an index for neonate survival using camera traps paired with decoys that mimic white-tailed deer (*Odocoileus virginianus*) neonates in the first month of life. We monitored sites with camera traps for two weeks before and after the placement of the neonate decoy and urine scent lure. Predator response to the decoy was classified into three categories: did not approach, approached within 2.5 m but did not touch the decoy, or physically touched the decoy; when conducting survival analyses, we considered these second two categories as dead neonates. The majority (76.3%) of the predators approached the decoy, with 51.1% initiating physical contact. Decoy probability of survival was 0.31 (95% CI = 0.22, 0.35) for a 30-day period. Decoys within the geographic range of American black bear (*Ursus americanus*) were primarily (75%) attacked by bears. Overall, neonate survival probability decreased as predator abundance increased. The camera-decoy protocol required about ½ the effort and 1/3 the budget of traditional capture-track approaches. We conclude that the camera-decoy approach is a cost-effective method to estimate a neonate survival probability index based on depredation probability and identify which predators are most important.

## Introduction

Newborn ungulates face the greatest mortality rates and are the most vulnerable to the risk of depredation within the first few weeks of life [1, 2]. Variation in survival from the depredation of these young animals can have a greater influence on population growth rates for a species

**Competing interests:** The authors have declared that no competing interests exist.

than other life stages [3, 4]. Areas with high predator abundance can result in lower survival of neonates, leading to the eventual decline of local deer recruitment rates [5]. Therefore, reliable measurements of neonate survival in relation to predator response are important for population dynamics models that inform adaptive management strategies. However, obtaining these measurements can be difficult as many ungulate neonates rely on cryptic coloration, smaller body sizes, and minimal movements to evade detection by predators [6].

Consequently, these characteristics likely impair researchers' ability to locate individuals of these hider species. The most common method to estimate survival for hider species of ungulates involves capturing neonates and then radio-tracking them to discover mortality events. This "capture-track approach" is expensive, time-consuming, and has potential drawbacks associated with invasive protocols. The approach requires the monitoring of pregnant does with vaginal implantation transmitters (VIT) that allow researchers to identify the exact time and place of birth [7]. This method requires extensive survey effort and supply costs to capture adult females (hereafter, does), monitor VIT signals, and track collared neonates. Alternative methods are to locate neonates opportunistically by following known pregnant does, spotlighting, or grid searches. Studies that use an opportunistic approach likely have estimates of survival biased high because captured neonates might be multiple days old, and most neonate mortality is within the first few days of life [2, 7]. Additionally, neonates that rely on a hiding strategy to evade predators may be difficult to find opportunistically, causing limitations on sample size [7]. Alternatives for estimating the survival for hider species that are cost-effective, require less survey effort, and are non-invasive would benefit wildlife managers in obtaining neonate survival estimates.

Catching and radio-collaring neonates is risky to animal health. Multiple studies noted problems from accidental neonate deaths or censored neonates due to method-caused issues, such as strangulation from legs getting caught in collars or loss of collar signal from technology malfunctions [2, 8, 9]. These problems may bias results and jeopardize the public's support for such research, as exemplified by the 2015 decision by the Governor of Minnesota to halt moose (*Alces alces*) neonate tracking after a series of study-related mortalities (State of Minnesota Executive Order 15–10, 2015). Additionally, many capture-track studies have difficulty identifying the species of predator responsible for mortalities, resulting in uncertainty of species-specific caused mortality ranging from 2.0%-32.4% [10, 11].

As depredation is the primary cause of death for hiding neonates (e.g., 88%) [10], a less expensive and non-invasive method would be useful to estimate an index for neonate survival based on depredation probability and species-specific depredation probabilities. For example, in cases where predator species removal has been deemed necessary to promote neonate survival, an index of depredation probability would be critical to determine which predator species should be removed and to measure the efficacy of this intervention. The primary predator of white-tailed deer (*Odocoileus virginianus*) neonates varies across the species' range, presumably depending on the local predator community. For example, in Pennsylvania, American black bear (*Ursus americanus*) is the most common predator of white-tailed deer neonates [12], but coyote (*Canis latrans*) was the leading cause of predator-induced mortality for neonates in studies in North Carolina and South Carolina [2, 9] although bears were not present at those study sites. As predator diversity and abundance changes from one site to another, or over time, generalizing results calculated for another study system may not reflect local predator pressures on neonate survival.

Pairing camera traps with neonate decoys is a potential cost-effective alternative to estimate an index of neonate survival based on predator-specific depredation probability rates. Researchers have used life-like animal decoys for a variety of purposes. For example, fake snakes were used to investigate chimpanzee response to potentially harmful animals [13]. Other studies used decoys to document how birds respond to warning colorations of various

predator species [14] and depredation of eggs in nests [15]. Decoys have been shown to generate similar responses as the actual threat [13, 15]. Decoys also allow the implementation of specific study designs to investigate *a priori* questions that are difficult to test when research is dependent on catching living neonates.

We tested the camera-decoy approach for studying white-tailed deer neonate survival, assuming any predator approach to within 2.5 m of a decoy as a "depredation" event as the predator would likely not miss the prey and the prey would be unable to escape at that distance. We used these events to estimate neonate survival. We set cameras in a stratified study design with locations throughout eastern North Carolina and in areas with varying levels of understory and canopy structure. In the state, American black bears have actively managed ranges, which resulted in half of our surveys being within and outside black bear range. When present, black bears are a large source of mortality for fawns [8, 12, 16]. Thus, we predicted that the presence of black bear would change which predator species had the highest relative depredation rates [8, 12, 16]. We predicted that predator-specific relative depredation rates are greatest for black bears, followed by coyote, and bobcat (*Lynx rufus*). Additionally, we predicted that greater understory cover would decrease depredation events (i.e., coming within 2.5 m of a decoy) and that the neonate survival probability would decrease with greater abundance of all possible predators. Finally, we compared the capture-track and camera-decoy approaches for estimating neonate survival in terms of survey effort, costs, and potential biases in the data.

## Methods

### Study area

We conducted our study in 18 counties in eastern North Carolina, USA (Fig 1). The study areas ranged across 2 ecoregions (piedmont and coastal plain) and included sites on private

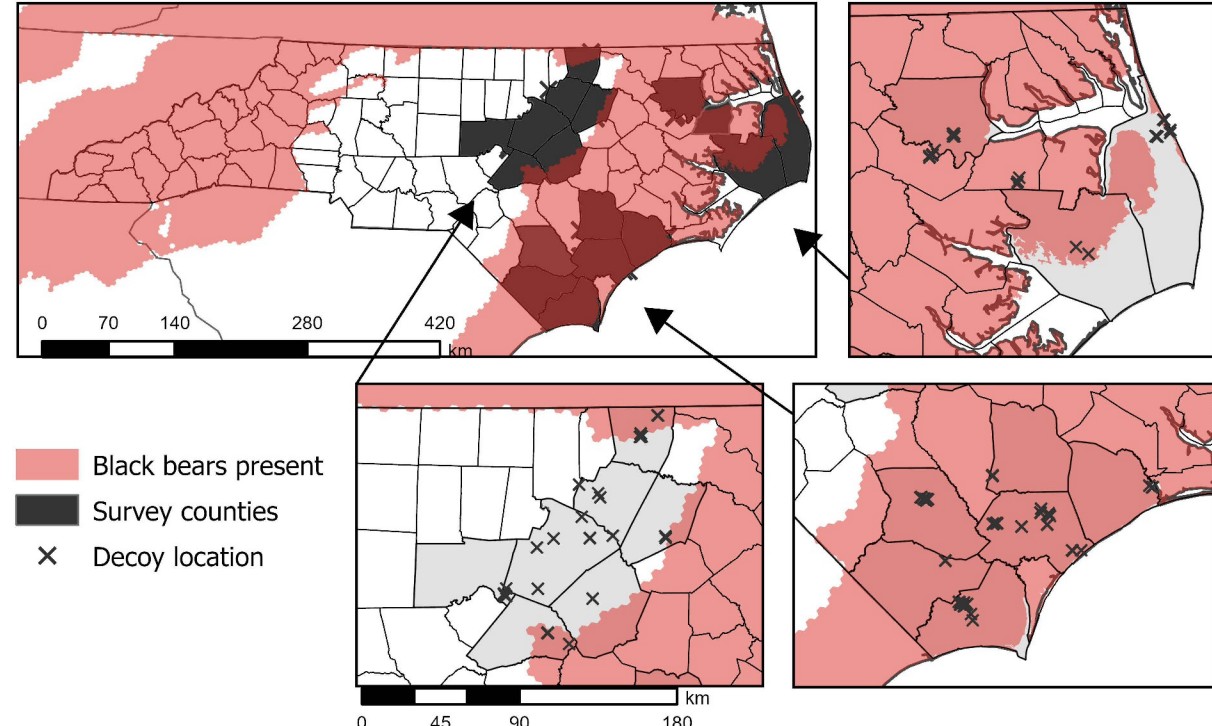

**Fig 1. Map of decoy and camera deployments in eastern North Carolina (2018).** X = Camera placements. The shaded area displays the IUCN black bear range map [18].

and public lands. White-tailed deer neonates are born from early-May to early July in North Carolina [17]. The coastal areas in North Carolina have the earliest estimated births, so we sampled first at the coast and moved west and inland over time. We timed decoy placements to match the peak of neonate births in each geographic area. North Carolina has 6 potential neonate predator species: black bear, coyote, bobcat, gray fox (*Urocyon cinereoargenteus)* [18], domestic dog (*Canis familiaris*), and black vulture (*Coragyps atratus*) [19]. All but black bear were potentially present at every location where cameras were placed; in the eastern half of North Carolina, black bear range is restricted to the coastal counties, such that 74 study locations were within bear range and 48 were not (Fig 1) [20].

To obtain variation in understory cover and tree basal area, we stratified camera locations across two land cover groups (open, forest) and along the transitional zone between them. Camera site locations placed in open areas were defined as having ≤25% of any combination of conifer and hardwood overstory. In each assigned geographic area, such as a state park or wildlife management areas, we placed camera set-ups ≥250 m apart and limited to no more than 8 locations per area to try to minimize any effect of an individual predator visiting multiple decoys. However, due to the predators having large home ranges, there may have been a small amount of non-independence at locations with more than one camera. Once an area was identified based on the above qualifications, we deployed cameras and conducted all vegetation surveys at the camera location.

## Camera and decoy surveys

We deployed camera-decoy experiments from May to July 2018. Each location had an initial 2-week survey period where 2 cameras were placed without the decoy or scent lure. This period was used to generate the site's predator relative abundance measurements before introducing the decoys. We used infrared Recoynx (RC55, PC800, and PC900; Recoynx, Inc., Holmen, Wisconsin) and Bushnell (Trophy Cam; Bushnell, Inc., Overland Park, Kansas) cameras traps. Both camera types were set to take still photos with a trigger speed of ≤0.5 seconds/trigger and 10 photos per camera trigger, at a rate of 1 frame per second. There was no delay period between triggers, and cameras would continue to trigger if an animal was still within the camera's scope. Cameras were set up ~40 cm above the ground (Fig 2). The 2 cameras were placed ~10 m from each other, slightly offset from facing one another (Fig 2).

After the initial camera survey, neonate decoys and a scent lure of neonate urine were deployed for 2 weeks at each camera location. We used an inflatable neonate decoy (Frantic Neonate Predator Decoy; Primos Hunting, Inc) that could be set up to look like it was a bedded neonate (Fig 2). Neonates are almost scentless, so we included only a small scent lure of neonate urine (The Predator bomb–Neonate Urine; Buck Bomb). We lightly sprayed a 12- x 20-cm piece of carpet with the scent lure in a plastic bag before going to the survey areas. The carpet scent lure and neonate decoy were placed in the center of the paired camera traps. The decoy was set up so it was lying on its stomach and was secured to a metal stake with a rubber band (Fig 2). No vegetation was cleared in front of the cameras or around the decoy; however, cameras were placed to maximize detection at individual sites.

## Vegetation surveys

We measured understory structure as a measure of concealment using a 1.5-m tall pole marked with 10-cm banded sections [21]. We recorded 5 measurements every 2.5 m in each of the 4 cardinal directions moving away from the decoy, which yielded a total of 20 pole readings. We recorded whether vegetation touched any of the 10-cm marked bands at each reading location. To get a measurement of understory structure, we divided the number of bands

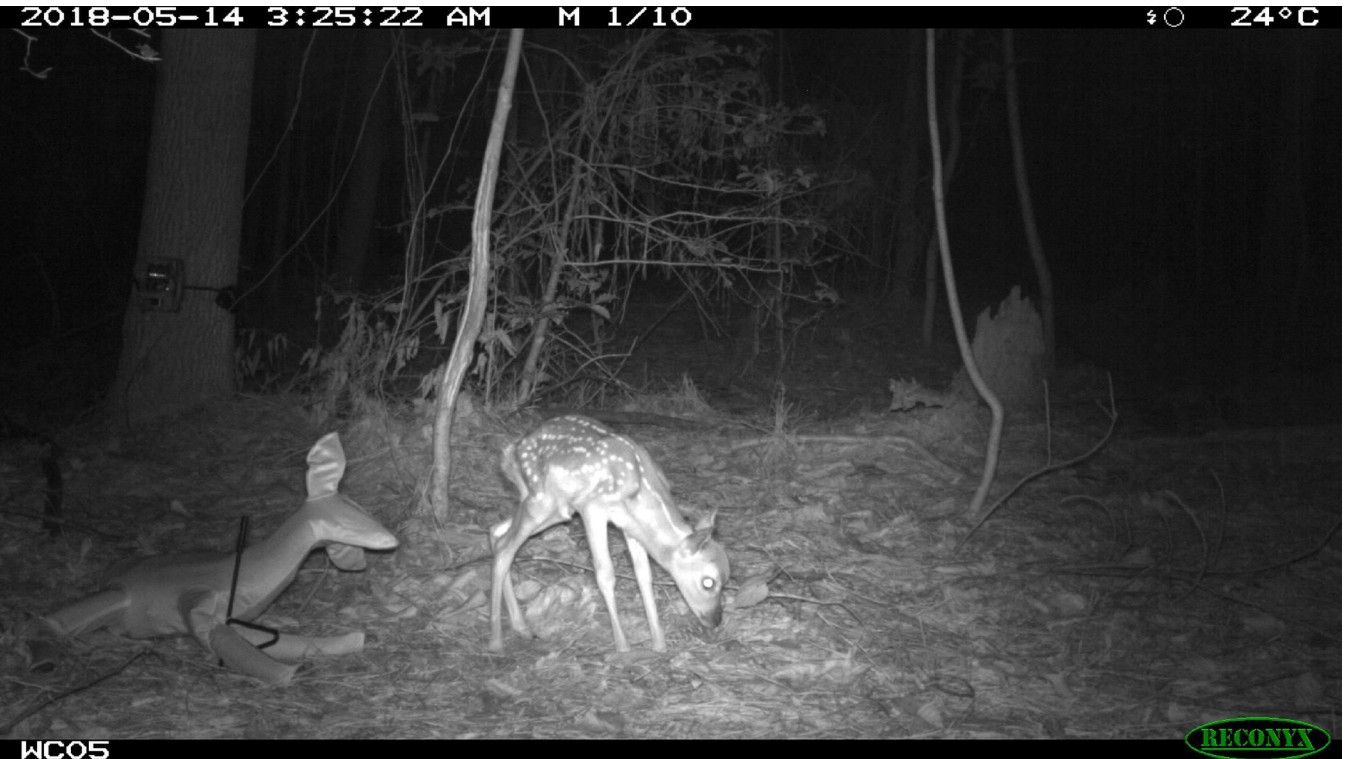

**Fig 2.** Camera trap photo showing the decoy (left) next to a living neonate (right) that happened to walk through our experiment. We also placed a bit of neonate urine on a 12- x 20-cm piece of carpet as a scent lure underneath decoy.

touched by vegetation by the total number of bands (i.e., 15 10-cm bands on each of the 20 pole readings, yielding 300 total bands). We measured canopy cover using a spherical densiometer. With the densiometer, we counted the number of gridded squares (96 possible sections) not occupied by canopy and multiplied the total by 1.04 to obtain a percent of the overhead area not occupied by the tree canopy. The difference between the unoccupied and 100 was used as the estimate of canopy cover. We measured basal area with a 10-factor prism, counting all trees that were "in" the circular plot. The number of "in" trees was multiplied by 10 to give a basal area estimate in $ft^2$/acre, which was converted to metric. To limit the human scent and potential alteration to vegetation structure, we collected the vegetation data at the retrieval of the camera and decoy for each site.

## Data classification and organization

The pre-decoy camera survey photographs were processed using the eMammal application [22]. The application combined photo detections into sequences if the images were taken <1 min apart. We used the pre-decoy camera survey to collect counts of all predator detections to be used for the predator relative abundance calculations. We manually processed the images taken when the decoy was deployed and maintained that photo detections were a sequence if the images were taken <1 min apart. We quantified predator response only for sequences taken during the decoy deployment. For each sequence with a predator detection, we recorded the specific species, the time of event, the time difference between that event and decoy set up, and number of individuals. Additionally, we classified each predator response to the decoy based on 3 classifications: 1. did not approach the decoy, 2. approached within 2.5 m in any

direction of the decoy with visual on decoy, and 3. physically touched the decoy. We considered the approached within 2.5 m and physically touched classifications to result in a neonate mortality because we assumed that a predator would spot a neonate at that distance and opportunistically attack the nearby neonate. When placing the decoy, we measured a 2.5 m radius around the decoy using flagging and triggered each camera to produce a reference image to assist in classifying predator response. Any site without any predators approaching the decoy within 2.5 m was classified as a surviving neonate.

## Predator-specific relative depredation rate

We calculated predator-specific relative depredation rates as the number of predator specific caused fawn "mortalities" divided by the total number of fawn "mortalities". We quantified predator-specific relative depredation rates in 2 different ways while using data collected only when decoys were deployed. First, we considered all "mortality" events (approached within 2.5 m and physical contact), including multiple 'mortalities' of the same decoy, in cases where the first predator did not physically destroy or move the decoy [15]. Second, to guard against the same individual repeatedly visiting the same decoy, we considered the relative depredation rate of each species when only the first predator detection across all species was considered per site [15]. This first measure is better for describing the full community of potential predators. whereas the second measure is more likely to give a depredation rate closer to what is experienced by a population of fawns.

## Covariates

To investigate the effect of the total predator community on decoy survival, we used an index of total predator relative abundance (i.e., number of individual predator counts / number of days camera trap deployed) calculated from the predator detections taken during the camera survey before the decoys were placed. Additionally, we used measures of understory structure, basal area, and canopy cover as the vegetation covariates. Vegetation types were identified by dominant overstory vegetation. Sites were classified based on 4 vegetation types: hardwood (≥75% hardwood overstory), conifer (≥75% conifer overstory), mixed (≥25% or ≤75% conifer and hardwood overstory), and open (<25% combination of conifer and hardwood overstory). Vegetation type was identified using the 2016 USGS National Land Cover Dataset (https://www.mrlc.gov/data, accessed January 2017). We included the distance in which the camera can detect an animal as a camera related factor that could affect the detection probability. Additionally, we identified whether a site was in black bear range using the ICUN American black bear range map [20].

All continuous data covariates were Z-scaled to ensure a standardized scale was used in all models. We examined Pearson's correlation coefficient for all the covariates and if any covariate pair had a value >-0.5 or <0.5, the covariate deemed most biologically relevant to neonate survival remained. We tested for multicollinearity using a variance inflation factor (VIF). Any models with VIF values >3 were not considered further.

## Daily neonate survival probability index

To model an index of daily neonate survival probabilities, we used a nest survival model in package *RMark* in Program R using "mortality" data collected during the decoy survey period [23–25]. The camera-decoy method meets the nest model assumptions: known number of days decoys placed in system, known decoy fate, final decoy check did not influence overall decoy survival, and decoy sites were independent from one another [23]. Nest survival models use a likelihood function that incorporates the number of days between checks, days until

**Table 1. Details of environmental covariates used in the analysis of neonate decoy survival in central and eastern North Carolina (2018).**

| Covariate | Details |
|---|---|
| **Predator presence** | |
| Predator relative abundance | Number of detections of all predators in the pre-survey period divided by the number of deployment days without a decoy. Predators used: black bear, bobcat, coyote, gray fox, and black vulture. |
| **Vegetation type classes** | |
| Mixed | Camera site locations placed in mixed forest characterized by both conifers and deciduous trees (>25% or <75% conifer and hardwood overstory) |
| Open | Camera site locations placed in open clearing (≤25% any combination of conifer and hardwood overstory) |
| Conifer | Camera site locations placed in forests dominated by conifer trees (≥75% conifer overstory) |
| Hardwood | Camera site locations placed in forest dominated by hardwood trees (≥75% hardwood overstory) |
| **Vegetation measurements** | |
| Basal area | Site tree basal area per acre |
| Canopy cover | Forest overstory density |
| Understory structure | Proportion of site occupied by vegetation within 1.5 m of the ground. |
| **Camera-related factors** | |
| Detection distance | The maximum detection distance that one camera was able to be triggered by a person at each site. |

depredation, and end fate to model daily survival probabilities [23]. For the survival model, we only used the first predator detection leading to a mortality event per site. In the model, we defined the *FirstFound* data column to reflect day 1 of a decoy survey. The number of days between the decoy set up and when the predator approached the decoy was recorded as *Last-Present* time estimate. We used the total number of decoy deployment days for the *LastChecked* required information. To code *Fate*, we assumed that sites with no predators detected or sites where no predators approached the decoys resulted in a sample that survived (*Fate* = 0). Sites where potential predators at least approached within 2.5 m of the decoy received a *Fate* = 1 representing mortality caused by depredation.

Survival models were calculated across all sites and separately for sites that were within and outside of black bear range. We ran the survival models with the selected *a priori* covariates (basal area, canopy cover, understory structure, vegetation type, and total predator relative abundance) by generating a global model and running all biologically reasonable potential model combinations using the dredge function from package *MuMIn* (Table 1) [26]. Model outputs were ranked using Akaike Information Criterion modified for small sample sizes (AICc). We made inferences of specific covariate effects based the parameter confidence intervals identified in the top competing models (ΔAICc ≤ 2.0) [27].

## Cost analysis

To compare overall costs and survey effort between capture-track and camera-decoy methods, we estimated the approximate costs associated with supplies and survey effort for both methods. We scaled the supply numbers and hours of survey to represent a sampling of 100 total neonates. For both methods, we assumed personnel hourly wage was $10. Supply items that can be used multiple times were recorded as reusable. The supply number and costs for the capture-track method were potentially underestimated as there are several ways to catch neonates, and supply malfunctions were not factored in. Additionally, depending on personnel

skills or the local abundance of deer, total study effort required to sample 100 neonates could change.

Supply costs were based on the item prices used during this study for the camera-decoy method. As multiple camera types are available for purchase and the cameras used were not purchased specifically for this study, we assumed a generic pricing of $200 per camera. Because camera traps are often reused between studies, we also provided an estimate without these purchase costs. For this survey, only one person was needed to run the camera-decoy method. However, depending on survey conditions, more than one individual should be considered to minimize safety concerns [28] and costs will be increased per individual added. Estimates of supply costs and survey effort for the capture-track method were based on the Chitwood et al. [2] survey design. The lead author of that paper was consulted to obtain approximate costs and numbers of selected supplies and man hours of effort. Based on the numbers discussed, survey effort and supplies were scaled to match the 100 neonate sample size estimates. For supply costs and overall effort, it was assumed that 1 vaginal implant transmitter resulted in 1 neonate. It was also assumed that no capture-related issues, such as loss of collar, occurred when scaling these costs, making our estimates conservative. We did not include DNA test costs for the capture-track method as some studies may not test carcasses or other remains for predator DNA.

## Results

We surveyed 122 camera locations from May to July 2018 for 1,798 pre-decoy survey days and 1,550 decoy survey days. We detected coyote, American black bear, bobcat, gray fox, black vulture, and turkey vulture (*Cathartes aura*) interacting directly and within 2.5 m of the decoys. We did not detect domestic dogs in either survey period, and the pre-survey did not detect black or turkey vultures. We did not consider turkey vultures as potential predators because they are purely scavengers [19]. Of predators, black bear had the highest average detection rates for both survey occasions (0.024/day SE = 0.01 and 0.035/day SE = 0.01; Fig 3). Bobcat had the lowest average detection rates (0.002/day SE = 0.00 and 0.004/day SE = 0.00; Fig 3). Excluding black vulture, the average detection rate for all predator species had error bars overlapping for pre and decoy surveys, indicating no change in predator detection with the decoy in place (Fig 3). This lack of difference before and after placement of the decoy and scent lure suggests that the decoy and lure may not strongly attract or repel predators that were not already present at the location.

During the decoy survey, 48 of 122 sites (39.3%) had at least one predator detection, totaling 131 predator detections. Of these detections, 31 (23.7%) predators did not approach the decoy, 33 (25.2%) approached within 2.5 m with visual but did not touch, and 67 (51.1%) physically touched the decoy. Black bear (85.1%) and bobcat (83.3%) had most of their responses lead to physical contact with the decoy (Fig 4). Across all sites, black bear accounted for 47% of the events considered mortality of the decoy (Table 2). Black bear occupied 61% of the survey sites that were in black bear range (45 out of 74 potential sites). When considering only sites within the geographic range of bears, black bears accounted for 74.6% of the events considered mortality of the decoy (Table 2). Coyote depredation rates were 18.9% at sites outside black bear's range and 4.8% inside black bear range (Table 2). Bobcat depredation rates were relatively similar in bear occupied and absent sites (6.4% vs 5.4%; Table 2).

Throughout the survey, 29 of 122 sites (23.8%) had multiple predator events. One site had 11 predator events by black bear. Eleven sites had two different species interact with the decoys. When only the first predator per site was considered, coyote and bobcat had a higher proportion of importance in the overall depredation rate, especially outside of the black bear range (Table 2). Many sites had repeated visits by the same species, especially from black bear,

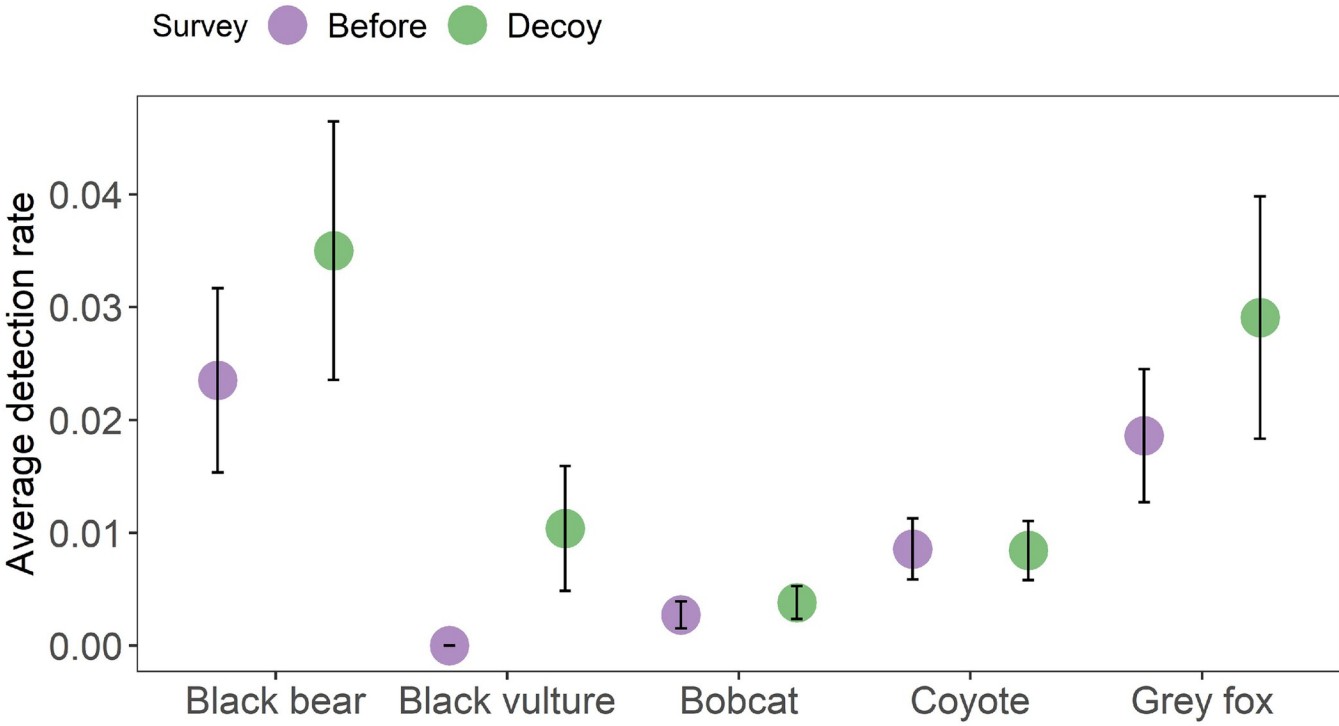

**Fig 3. Average detection rates as relative abundance for 5 predators detected during a 2-week pre-survey and a 2-week period with a neonate decoy in place (2018).** Error bars = standard error. Detection Rate = Number of species detections at a survey / survey nights.

black vulture, and gray fox, may have inflated the overall species-specific relative depredation rates.

The survival analysis using only the first depredation event had 14 competing models with a ΔAICc ≤ 2.0 (Table 3). The model survival probability estimates fell between 0.31 (95% CI = 0.21,0.44) and 0.32 (95% CI = 0.21, 0.44) over 30 days. Across all competing models, only the pre-survey total predator relative abundance negatively affected decoy survival and the only variable present in all models (Table 3).

The supplies and survey effort to run a decoy-camera survey were ~32.3% of the costs to run a capture-track survey (S1 Table). To get a sample equivalent of 100 neonates, the approximate supply totals for a capture-track survey were $77,170 compared to camera-decoy costs of $38,063 (S1 Table). As capture-track requires capturing pregnant does before parturition and neonates after parturition, survey effort costs of the capture-track method also greatly exceed that of the camera-decoy method that operates over a shorter time period (respectively, $40,500 vs. $13,440, S1 Table). The survey effort to run the camera-decoy protocol requires only ~45% of the number of survey hours necessary to run a capture-track survey (S1 Table). Unlike in the capture-track method, the majority of the supplies in camera-decoy surveys are reusable or useable for other sampling surveys (S1 Table). Indeed, if the camera-decoy study is done with reused cameras, the price drops to $18,063, or ~15% of the capture-track approach. The cost difference between the camera-decoy study would be even greater if researchers used GPS collars for their capture-track studies instead of the VHF collars used to estimate costs.

The camera-decoy method produced a similar probability of survival for white-tailed neonates to what is reported in the literature, especially in the southeastern USA (S2 Table). Predator specific relative depredation rates from this study were similar to other studies for bobcats, coyotes, and black bears but higher for black vultures and gray fox (S2 Table).

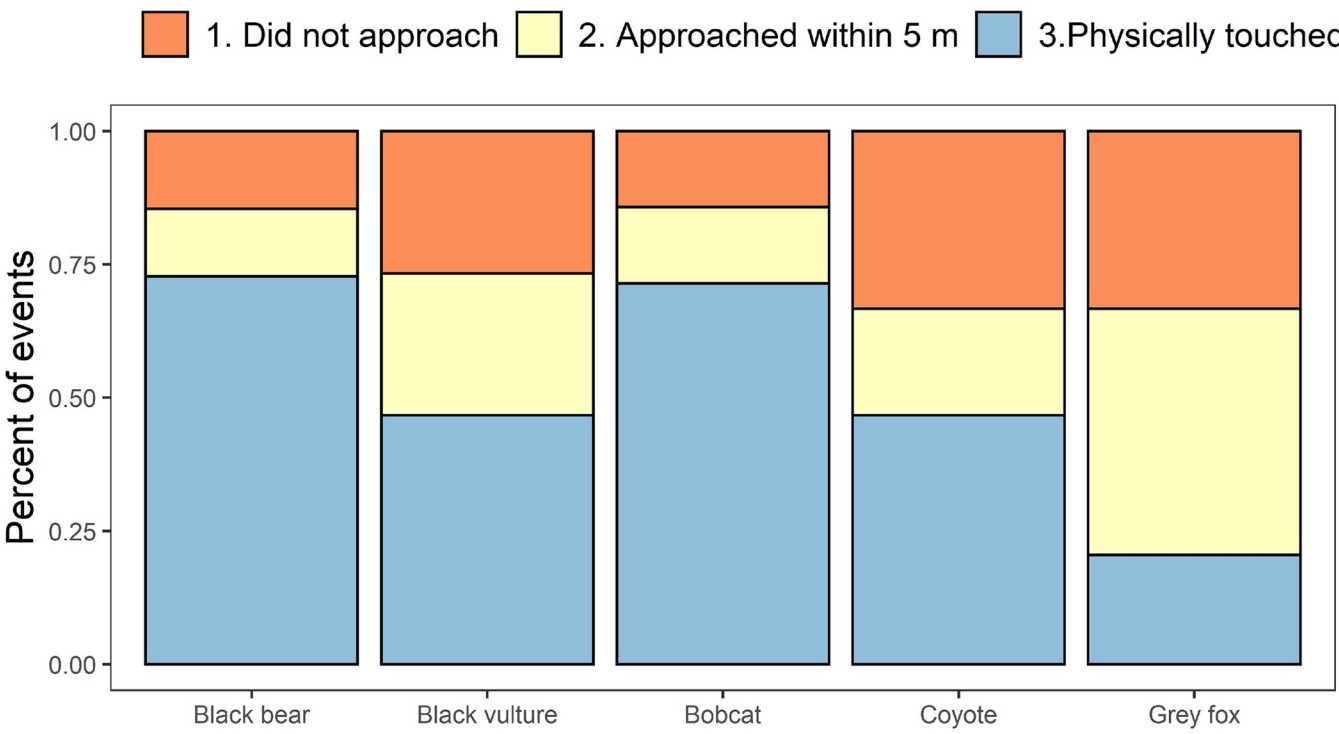

**Fig 4. Summary of total predator response to the neonate decoys in central and eastern North Carolina (2018).** Numbers include multiple encounters per site. We considered only approached within 2.5 m and physically touched (Blue) classifications as depredation events.

## Discussion

We determined that the camera-decoy method has lower overall costs than the capture-track method and produces similar results on neonate survival probabilities and predator-specific relative depredation rates as reported in studies using traditional methods. All of the potential predator species in our study responded to the decoys, with the majority of interactions being

**Table 2. Summary of neonate depredation events based on species group and black bear range in North Carolina (2018).** Count = the number of detections where an animal at least approached within 2.5 m of the decoy. Numbers do not include species detections that did not approach the decoys. Based on 83 predator responses that either approached within 2.5 meters or approached and physically touched the decoys. If more than one predator was present in a capture sequence, only one predator was considered towards the % of depredation estimates.

| | | All sites | | Within black bear range | | Outside black bear range | |
|---|---|---|---|---|---|---|---|
| | # of Sites | 122 | | 74 | | 48 | |
| | **Species** | **% of depredation** | **Count** | **% of depredation** | **Count** | **% of depredation** | **Count** |
| All predator detections | Black bear | 47.0 | 47 | 74.6 | 47 | 0 | 0 |
| | Coyote | 10.0 | 10 | 4.8 | 3 | 18.9 | 7 |
| | Bobcat | 6.0 | 6 | 6.4 | 4 | 5.4 | 2 |
| | Gray fox | 26.0 | 26 | 12.7 | 8 | 48.6 | 18 |
| | Black vulture | 11.0 | 11 | 1.6 | 1 | 27.0 | 10 |
| Only first predator detection per site | Black bear | 43.9 | 18 | 66.7 | 18 | 0 | 0 |
| | Coyote | 17.1 | 7 | 7.4 | 2 | 35.7 | 5 |
| | Bobcat | 7.3 | 3 | 3.7 | 1 | 14.3 | 2 |
| | Gray fox | 24.4 | 10 | 22.2 | 6 | 28.6 | 4 |
| | Black vulture | 4.9 | 2 | 0 | 0 | 15.4 | 2 |

**Table 3. Top competing model (ΔAICc ≤ 2.0) outputs for neonate decoy survival for all sites in North Carolina (2018).** S = survival probability. Table includes linear predictors and 95% confidence intervals for parameter estimates. ΔAICc = The difference in AICc from the AICc score of the model with the relative best fit. df = degrees of freedom. logLik = log likelihood. Weight = influence of observations on model parameters.

| Linear Predictors and 95% CIs for survival models | ΔAICc | df | logLik | Weight |
|---|---|---|---|---|
| Predator Relative Abundance (-0.72, -0.09) + Basal Area (-0.01, 0.72) | 0 | 3 | -89.01 | 0.12 |
| Predator Relative Abundance (-0.75, -0.12) + Canopy Cover (0.01, 0.60) | 0.04 | 3 | -89.03 | 0.12 |
| Predator Relative Abundance (-0.73, -0.10) + Basal Area (-0.16, 0.63) + Canopy Cover (-0.13, 0.54) | 0.58 | 4 | -88.29 | 0.09 |
| Predator Relative Abundance (-0.79, -0.14) + Canopy Cover (-0.03, 0.57) + Understory Structure (-0.44, 0.12) | 0.88 | 4 | -88.44 | 0.08 |
| Predator Relative Abundance (-0.77, -0.11) + Basal Area (-0.06, 0.69)+ Understory Structure (-0.45, 0.13) | 0.93 | 4 | -88.47 | 0.08 |
| Predator Relative Abundance (-0.76, -0.11) + Basal Area (-0.01, 0.72) + Conifer (-1.11, 0.33) | 0.96 | 4 | -88.48 | 0.08 |
| Predator Relative Abundance (-0.73, -0.10) + Basal Area (0.01, 0.75) + Hardwood (-0.43, 1.20) | 1.10 | 4 | -88.55 | 0.07 |
| Predator Relative Abundance (-0.77, -0.11) + Basal Area (-0.19, 0.61) + Canopy Cover (-0.15, 0.52) + Understory Structure (-0.42, 0.15) | 1.73 | 5 | -87.86 | 0.07 |
| Predator Relative Abundance (-0.71, -0.09) + Basal Area (-0.07, 0.70) + Open (-1.12, 0.64) | 1.74 | 4 | -88.87 | 0.05 |
| Predator Relative Abundance (-0.81, -0.16) + Understory Structure (-0.50, 0.07) | 1.77 | 3 | -89.89 | 0.05 |
| Predator Relative Abundance (-0.73, -0.09) + Basal Area (-0.08, 0.70) + Mixed (-0.54, 0.89) | 1.78 | 4 | -88.89 | 0.05 |
| Predator Relative Abundance (-0.74, -0.12) + Canopy Cover (-0.05, 0.59) + Mixed (-0.52, 0.89) | 1.78 | 4 | -88.89 | 0.05 |
| Predator Relative Abundance (-0.76, -0.12) + Canopy Cover (-0.02, 0.59) + Conifer (-0.92, 0.56) | 1.84 | 4 | -88.92 | 0.05 |
| Predator Relative Abundance (-0.80, -0.13) + Basal Area (-0.05, 0.69) + Understory Structure (-0.46, 0.13) + Conifer (-1.12, 0.33) | 1.87 | 5 | -87.92 | 0.05 |

classified as leading to decoy mortality, suggesting predators treated the decoy as a real neonate. Using decoys and cameras also allows the implementation of an *a-priori* study design that is non-invasive, cost-effective, and can be replicated easily, enabling larger sample sizes. Furthermore, the camera-decoy approach allows predators to be identified with certainty and can provide a measure of relative abundance of the entire predator community.

Although the camera-decoy method only accounts for the depredation portion of neonate survival (i.e., not disease, starvation, or abandonment), our index of a 31% decoy survival was similar to estimates generated from capture-track methods throughout the southeastern United States, where the majority of fawn mortality is due to depredation within the first 30 days of life (e.g., 33% [29], 21% [9], 29% [30], 27% [16]). Black bear had greater relative depredation rates than any other species in our study, which is a similar result to studies with black bears present [16, 31, 32]. Though not every study has documented black bear as the top predator, the majority of studies have shown black bear at least being the second most impactful predator [8, 12]. Neonate mortality studies that do not have black bear in their system often identify coyote and bobcat as the most important predator. We documented similar depredation rates for areas where coyotes and bobcat were the top predator [2, 9, 11]. Additionally, using decoys to identify depredation of neonates can be useful as it identifies areas with high indexes of depredation rates while collecting information on relative abundance of predators. This information can identify which predators significantly influence neonate survival and assist managers in managing predation risk.

The ability to place decoys using an *a-priori* study design allowed us to investigate the differences in predator relative depredation rates while also testing for vegetation effects on survival across sites. By not having to rely on living neonates, the camera-decoy method can enable systematic *a-priori* study design and the testing of research and management hypotheses. Additionally, because the camera-decoy method requires less expense and survey effort, larger sample sizes can be reached over larger spatial areas than the capture-track method. Currently, no capture-track study has surveyed neonate survival at as large of scale (16 counties) as our decoy study. This suggests that the camera-decoy method could provide a way to expand the spatial areas of neonate survival studies and have the ability to set up designs to investigate new questions about environmental effects on neonate survival over larger scales. Moreover, survival estimates over a larger spatial area and over a longer time can generate more relevant population estimates of the overall deer herd than using probabilities from a much smaller scale.

We showed that the relative abundance of predators had a negative effect on neonate survival probabilities, which is similar to what other studies have documented [8, 16]. The pre-decoy portion of our survey allowed for an index of predator relative abundance to be incorporated into survival models. The sites outside black bear range that were in mixed forest only had 4 out of the 26 sites have any detections of predators, suggesting that this forest type may have lower depredation pressure. Other studies similarly have reported that vegetation characteristics at a bed-site had either a weak or no effect on neonate survival [2, 9].

The similarities in the survival estimates between this study and other neonate survival studies indicate the camera-decoy method can be a useful non-invasive alternative to capture-track methods. Future research needs to be conducted to see how camera-decoy method generated survival rates reflect survival generated from traditional capture-track approaches. We compared our survival estimate to other studies that mostly were conducted in the southeastern USA, including one conducted in a county that borders the areas we surveyed [2]. To best compare the capture-track and decoy-methods would require use of both methods concurrently in the same general location. Unlike capture-track methods, the decoy-method does not account for neonates that die from causes other than predators, such as illness, abandonment, or starvation. Additionally, a survival index based on depredation probability does not consider maternal defense, maternal movement, and older fawn mobility (>10 days) [2]. Maternal investment, such as direct protection and the selection of the fawn bedding sites, may influence the survival of fawns [33, 34]. However, we still believe our approach is useful because depredation typically makes up most of the total neonate mortality (e.g., 88%) [16] and might be even higher given the risk of study related effects in capture-track studies. Although the decoy method cannot incorporate maternal investment, information on local maternal bedding site selection can help inform decoy placement in the most appropriate vegetation conditions. Another potential source of error in the camera-decoy method is our assumption that all predators approaching within 2.5 m will successfully kill the neonate. It is possible that some older neonates would be able to escape from some predators (e.g., black vulture). Lastly, more testing needs to be performed to test the effect of the scent lure and decoy in attracting predators from outside the local area.

The success of decoys for surveying neonate mortality, and the relative ease and low expense compared to traditional methods, opens up several new research questions that could be addressed by this method. First, we recommend a synchronized neonate mortality study using capture-track and camera-decoy methodologies at the same site to compare differences in survival estimates with local site information on depredation and starvation caused neonate mortalities. Second, we believe the camera-decoy mortality studies should be replicated across space and time to document the spatial and temporal variation in relative depredation rate and

its relationship to local predator abundance. Third, further study on the use of predator specific effects such as presence/density and time to detection would be a next step into comparing the camera-decoy method with capture-track methods as predator densities can influence predation rates [9]. Understanding this variation could help parameterize integrated population models and reduce the uncertainty of their predictions [35]. Finally, we suggest there is a great potential to vary the combination of visual, olfactory, and auditory cues in camera-decoy studies to experimentally study the senses used by different predators to locate prey. Although further research is needed, the camera-decoy method shows promise in being a non-invasive and cost-effective option to estimate indices of fawn survival and predator relative depredation rates.

## Supporting information

**S1 Table. Cost analysis comparing a typical live capture neonate project design vs. a camera trap survey paired with decoys and scent lures.**
(DOCX)

**S2 Table. White-tailed deer neonate survival, predator species-specific rates of depredation, and rate of uncertainty in mortality for studies of fawn survival across North America.**
(DOCX)

**S1 File. Fawn decoy "mortality" events and corresponding covariates dataset.**
(XLSX)

## Acknowledgments

We thank the private landowners and North Carolina Wildlife Resources Commission biologists who allowed us to access private and public lands, respectively. We thank E. Briggs for assistance checking video footage for classifications. M. Chitwood provided information on costs and survey effort for running a capture-track project for neonates based the Chitwood et al. (2015) study. Finally, the North Carolina Museum of Natural Sciences and North Carolina State University Department of Forestry and Environmental Resources provided critical resources.

## Author Contributions

**Conceptualization:** Hailey M. Boone, Krishna Pacifici, Christopher E. Moorman.

**Data curation:** Hailey M. Boone.

**Formal analysis:** Hailey M. Boone.

**Funding acquisition:** Roland Kays.

**Investigation:** Hailey M. Boone.

**Methodology:** Hailey M. Boone, Krishna Pacifici, Christopher E. Moorman, Roland Kays.

**Resources:** Roland Kays.

**Supervision:** Roland Kays.

**Visualization:** Hailey M. Boone.

**Writing – original draft:** Hailey M. Boone.

**Writing – review & editing:** Hailey M. Boone, Krishna Pacifici, Christopher E. Moorman, Roland Kays.

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
