## [Decision Letter · Decision Letter 0]

1 Aug 2023

PONE-D-23-13153Using decoys and camera traps to estimate depredation rates and neonate survivalPLOS ONE

Dear Dr. Boone,

Thank you for submitting your manuscript to PLOS ONE. After careful consideration, we feel that it has merit but does not fully meet PLOS ONE’s publication criteria as it currently stands. Therefore, we invite you to submit a revised version of the manuscript that addresses the points raised during the review process.

Both reviewers have highlighted important issues in your manuscript.

Reviewer 1 indicated that you must conduct a proper test to compare with the other study conducted in the same area at the same time. Also, you should consider that predation rates may be affected by mother decisions that could be difficult to reproduce, then that must be discussed.  

Reviewer 2 has similar issues. This reviewer emphasizes that one problem is how the analysis was described in the methods section, but also that you should consider for uncertainty in predators’ presence and their activities.

We look forward to receiving your revised manuscript.

Kind regards,

Paulo Corti, Ph.D.

Academic Editor

PLOS ONE

3. We note that Figure 2 in your submission contain copyrighted images. All PLOS content is published under the Creative Commons Attribution License (CC BY 4.0), which means that the manuscript, images, and Supporting Information files will be freely available online, and any third party is permitted to access, download, copy, distribute, and use these materials in any way, even commercially, with proper attribution. For more information, see our copyright guidelines: http://journals.plos.org/plosone/s/licenses-and-copyright.

1. You may seek permission from the original copyright holder of Figure 2 to publish the content specifically under the CC BY 4.0 license.

Reviewers' comments:

Reviewer's Responses to Questions

**Comments to the Author**

1. Is the manuscript technically sound, and do the data support the conclusions?

Reviewer #1: Partly

Reviewer #2: Partly

2. Has the statistical analysis been performed appropriately and rigorously? 

Reviewer #1: Yes

Reviewer #2: No

3. Have the authors made all data underlying the findings in their manuscript fully available?

Reviewer #1: Yes

Reviewer #2: No

4. Is the manuscript presented in an intelligible fashion and written in standard English?

Reviewer #1: Yes

Reviewer #2: Yes

5. Review Comments to the Author

Reviewer #1: This manuscript presents an interesting approach to quantify the percentage of ungulate neonate mortality determined by predation, based on experimental decoys and neonate odor.

I have no technical issue to raise. However, I have two major comments. First, authors have found that the potential estimate of mortality was comparable to those obtained through other studies based on the tracking of collared individuals. Although I am confident that the results of this comparison are reasonable, a proper test should be based on the comparison with a study conducted on the same population at the same time. This task would be clearly difficult to accomplish, but I think it should be acknowledged by authors in the Discussion also , describing the ecological conditions of the other studies used for comparisons.

Second, it may be possible that natural predation rates be affected by ungulate selection of concealed sites for bedding/hiding neonates. The deployment of decoys by observers is assumed to mimic the selection of bedding/hiding sites by ungulate mothers or neonates, which is hard to verify and which may influence comparisons. This issue should be considered and discussed thoroughly by authors.

As a minor comment, Tables 4-5 should include standardized model weights.

Reviewer #2: # Review for:

*Using decoys and camera traps to estimate depredation rates and neonate survival*

NOTE: I've written these comments in markdown format. I've also attached a pdf if that is easier for you to read.

In this paper the authors were interested in evaluating a new method to estimate neonate survival using camera traps. The paper has a logical flow to it, which makes it easier to understand, though some of the paragraphs could use some restructuring to improve flow a little more (e.g., the introduction). I thought that the suggested method was interesting, and the authors could even go a bit further to sell the fact that this technique is non-invasive (brought up a few times at the start but really only referenced quickly again in the discussion). However, I did have some concerns with the analysis. The biggest issue is how the analysis was described in the methods (see below), but also I wonder why the authors did not account for or at least explain why they did not account for uncertainty in predator presence on the landscape and also what the relative predator activity metric represents ecologically. More specifically:

1. The reporting of the methods leaves a lot of uncertainty with respect to how the models were fit, and what data was used. For example, it is still unclear to me if there was a pre-decoy and post decoy model fitted to the data (i.e., separate models for each species) or if the authors included a 'decoy' dummy variable. This is especially important to cover because the authors try to make inference about a decoy effect, or rather that there is none, and I cannot determine how their model actually did this based on the covariates they said they included. This makes it so that a reader would not likely be able to recreate the authors analysis from the information presented, which is a problem.

2. It is possible to account for uncertainty in predator presence on the landscape while fitting your survival models. There are a few way you could do this. First, if you are interested in predator specific effects, you could just fit an occupancy model with a time to detection observation model. This is something covered in depth (model and code) by Kéry & Royle (2015) and also something I used in Fidino et al. (2020, with data and code [here](https://github.com/mfidino/lure_project)). Conversely, you could fit standard occupancy models and then link that with the survival model. For a frequentist analysis, this could be done by using a bootstrap method and estimating predator occupancy at each site (leaving you a distribution of parameters for that term). For a Bayesian analysis you could just link the models together and fit the models at the same time (i.e., link a multi-species occupancy model with a survival model). This could, for example, allow you to estimate 1) the effect of multiple predators being present by summing across the predator incidence matrix or 2) the effect of individual predators with dummy variable. Finally, with a Bayesian analysis you could even use a Poisson distribution for the observation model, treating the number of times a species was detected (e.g., photos) as the response variable. By summing that across your predators you could have a 'detection corrected' relative abundance metric. And then from there you could link your a Bayesian survival model to this third type of model. Basically, because there are multiple avenues available to the authors, it strikes me as odd to assume no observational error occurs during data collection, which is what they are currently doing. I'm not encouraging a full reanalysis here, but some justification of their baseline assumptions they are making would help. For example, one relatively 'simple' way to justify their assumption is to fit some occupancy models with their predator data to estimate predator detection probabilities. The authors could then calculate the probability each predator is detected at least once with the number of survey days you have available (1 - (1 - p)^j where j is the number of days sampled). This information could then be shared so that the reader has an idea about how much each species may be contributing to the predator relative abundance metric, and if there are some species who are regularly present or absent from that covariate.

```

Fidino, M., Barnas, G. R., Lehrer, E. W., Murray, M. H., & Magle, S. B. (2020). Effect of lure on detecting mammals with camera traps. Wildlife Society Bulletin, 44(3), 543-552.

Kéry,M.,and J.A.Royle.2015. Applied hierarchical modeling in ecology: analysis of distribution, abundance and species richness in R and BUGS. AcademicPress,London,England,UnitedKingdom.

```

3. The predator relative abundance metric comes out as important in all of your top models. However, I am having a hard time trying to interpret what that would actually mean with respect to neonate survival. To me, it would seem as if each predator would vary in how successful they are with finding your decoy, and this metric assumes that predators are essentially exchangeable. Some more justification for this metric would help, and also some guidelines for how this should be interpreted ecologically. For example, the authors bring up how traditional methods make it difficult to determine the predator, while camera traps do make it possible. To me, that seems to indicate an interest in understanding predator specific effects, which this predator relative abundance metric does not capture.

I think that the authors can get passed a lot of these issues with some additional explanation / logic, which would not require a full reanalysis. Certainly, if the authors want to reanalyze their data to account for such issues they could, but at a minimum they would need to provide some caveats about the current statistical approach (and even indicate that these caveats could be solved with different styles of analysis).

If the authors have any specific questions for me about this review I can be reached at mfidino@lpzoo.org. I have section specific comments below, I hope the authors find them helpful.

Cheers,

Mason Fidino

## Abstract

---

### Line by line comments

line 26: To split up the commas a bit it would be good to use em dashes for `individuals less than four weeks old`. 100% my own writing style creeping in here, so feel free to ignore!

Line 27: `population dynamics of a population` is redundant. Just say "...can influence ungulate population dynamics" or something like that.

Line 39: You can definitely report probabilities at percentages like this, but when you do that it gets a little weird when you need to report uncertainty (e.g., 95% confidence intervals). For example, 31% (95% CI = 22%,35%) looks off given the multiple percentages. Instead, 0.31 (95% CI = 0.22, 0.35) looks much cleaner (and even saves space).

## Introduction

---

### Top-level thoughts

1. The first paragraph could probably get split into two. The first half is about neonate survival in ungulates while the second half is about how researchers estimate neonate survival. As such, the topic sentence does not really line up with half the content of this paragraph. You could get around this by adding a closing sentence after line 56 and then have the 'consequently' sentence there be the topic sentence of a new second paragraph. Further, what is currently paragraph 2 in this draft has a mix of redundant information and then more specificity than what is currently the second half of paragraph 1. If you split this paragraph you can fold in a lot of info from the current second paragraph / remove redundancies.

2. Making predictions with the 'average' model in this case is not good because the nest survival model does not use the identity link (at least in terms of calculating average parameters and then using those for predictions). This is covered at length by `Cade (2015)`. However, `Banner and Higgs (2017)` have some pointers on how this may actually be done. Chances are, if that authors used the standard approaches available, they first generated average parameter values and then used them for prediction. If that is the case, that is not the correct thing to do with this model (I believe you can model average predictions instead). Furthermore, the notion that you can get an idea about the 'overall effect' of a given parameter via model averaging is not really correct, and this error is unfortunately quite pervasive in our field. You might instead consider that you have quite a few uninformative parameters in your competing models (e.g., if they are nested subsets of one another), and if that is the case, you may just need to work off a single model (see Arnold 2010).

```

Arnold, T. W. (2010). Uninformative parameters and model selection using Akaike's Information Criterion. The Journal of Wildlife Management, 74(6), 1175-1178.

Banner, K. M., & Higgs, M. D. (2017). Considerations for assessing model averaging of regression coefficients. Ecological Applications, 27(1), 78-93.

Cade, B. S. (2015). Model averaging and muddled multimodel inferences. Ecology, 96(9), 2370-2382.

```

### Line by line comments

line 52: ...can have **a** greater influence...

Line 53-54: This sentence comes a bit out of left field. If variation in neonate survival is important, how does having reliable measurements of this variation inform adaptive management? Is it to control the variation? To increase survival? A little more clarity here would help to make sure the reader follows your logic. I don't disagree with your statement, but right now you are forcing the reader to fill in some logical gaps in order to agree with this.

Line 56: You can drop the 'also'

Line 63 - 64: This sentence is redundant with info from the previous paragraph.

Line 67: Change 'An alternative method' to 'Alternative methods' as you list multiple methods here.

Line 72: Missing a closing sentence on this paragraph.

Line 82 - 83: `...a cheaper and less intrusive method...`. Right now there is uncertainty if 'less' should be applied to the word 'intrusive.'

Line 95 - 102: Since the topic here is predation it could also make sense to cite some paper that used decoy eggs for nest predation.

Line 104: considering or assuming? You likely also need to defend this assumption / provide some logic behind it. Two come to mind to be: as 'hider' species, if a predator gets within that distance they are 1) not likely to miss the prey and 2) the prey lacks a strategy to escape the predator at that distance.

Line 110: You've been past tense with this paragraph so it would read better to say 'we predicted that depredation rates...' Also, why did you make this prediction?

## Methods

---

### Top-level thoughts

1. Why make predictions for only three predators when more are present?

2. If the goal was to limit an individual predator visiting multiple camera trapping locations the clustered design only really achieves that at the 'among-cluster' scale for many of these species (i.e., 250 m is still quite close for larger ranging species like black bear or coyote). This may cause some small amount of non-independence among camera trapping locations, which at a minimum should be acknowledged here.

3. Figure 2 was so small in the review I could not see it, but I'll take your word that the image gets the point across.

4. Your analysis does not account for the clustered nature of your study design. A cluster-based random effect would do this, but is unfortunately not available in `RMark`.

5. One thing that is not made clear in the methods is how the effect of neonate was quantified. I did not see any explanation about whether the first two weeks of data was used in the analysis or if that was simply done for acclimation. To me, it would seem like you could estimate weekly time to detection with all the data and include some binary variable to indicate a predation event (though there is some complication about what you would consider a predation event when the neonate decoy is not present). Regardless, some extra information here is welcome (note: looking at the results it seems like models were fit to each period seperately, not seeing where this is brought up in the methods though I could have missed it).

6. Using a single person for camera trap surveys makes the assumption that this is safe to do so, which is not always the case and may not consider the risks that marginalized communities face in our field (Rudzki et al. 2022 and references therein). For example, currently the camera trap method only requires 45% of the number of survey hours as the other method (line 350). Does that mean if two people were required it would be 90%? It could help to recognize that multiple people may be necessary depending on how safe it is to conduct fieldwork.

```

Rudzki, E. N., Kuebbing, S. E., Clark, D. R., Gharaibeh, B., Janecka, M. J., Kramp, R., ... & Richards‐Zawacki, C. L. (2022). A guide for developing a field research safety manual that explicitly considers risks for marginalized identities in the sciences. Methods in Ecology and Evolution, 13(11), 2318-2330.

```

### Line by line comments

Line 133 - 134: How was the proportion of conifer / hardwood overstory quantified?

Line 165: You say 1 dm here than 10-cm everywhere else. Use one or the other (though they are equivalent).

Line 184 - 190: How did you measure 2.5 meters from the neonate in the videos?

Line 201 - 211: How was this measured (was not covered in the methods so far).

Line 232 - 233: Why fit the models seperately for inside and outside black bear range? Could you have included a dummy variable there for black bear (plus interactions with other covariates) instead? You'd have the same number of parameters to estimate, but you'd be sharing information for other species which could leave to more precise estimates.

Line 233 - 235: So are we assuming that all possible combinations are biologically reasonable?

## Results

---

### Top-level thoughts

1. What was the naive occupancy of these different predators across the sampled areas? Right now your model assumes that, aside from the black bear, these species are available to prey on your decoy (or more specifically that they are present in each location).

### Line by line comments

Line 271 & 272: The reported detection rates are a little confusing because the second ones (0.035 & 0.004) lack the `/day` component.

Line 272 - 276: This was not covered in the methods, but did you fit the models separately (a pre-decoy model and a post decoy model)? If you did, you did not actually make any statistical comparison about a decoy effect here, meaning this sentence is not correct. Second, if you did quantify this effect, failing to detect an effect does not mean that there is no effect (i.e., you should not say there was no change). Instead, you failed to detect a difference. But again, if these were separate models you did not quantify a difference.

Line 280 - 290: How were all these depredation probabilities calculated? Given the lack of uncertainty in the estimates, it seems like these are raw summaries of the data? However, if that is the case, these estimates would be biased as they do not account for the fact that your data are right-censored (which is what your nest survival model accounts for). These depredation probabilities can be made from your model, and as such could also come with an uncertainty estimate. For example, you could calculate the cumulative probability that a neonate gets predated over the course of your study for each species (or even the cumulative probability a neonate gets predated by any species, perhaps inside and outside of black bear range).

## Discussion

---

### Top-level thoughts

1. One thing that is not really brought up is that there is no doubt spatial variation in deer across the landscape. How are camera trapping sites selected relative to this? Are you only sampling in areas where deer are known to occupy with a high likelihood?

2. The concluding paragraph lists some future directions but does not really bring us back to the contribution provided by this research. I'd circle back at the end to remind the reader about the key takeaways here / why this research helps move this field forward.

3. Do the authors have any tips for people who are interested in using this method?

### Line by line comments

Line 369 - 370: How is the camera-decoy approach not perfect (i.e., give an example or two to the reader about this).

Line 387 - 391: Again, I'm not sure that the authors specifically evaluated this effect given that they fit separate models to different subsets of the data, which means that they did not statistically evaluate / quantify a difference. Similarly, the authors calculated species-specific depredation rates from the raw data when it could have been estimated in the model. As such, the authors could make descriptive comparisons about their sample but it is a bit more difficult to assume that these value are representative of the larger population (i.e., using inferential statistics to get said estimates). As this paper is more methodological, some care should be taken here to not oversell what this technique can provide. For example, would it be necessary to get a much larger sample size to estimate predator specific effects? If so, what do the authors suggest?

## Tables & figures

---

ALL TABLES: Numeric columns should be right aligned and text columns should be left aligned. Right now it looks like the numeric columns are center aligned.

Table 3 - 5. Saying S = survival probability is a little odd. Looking at the models it appears to be a function in that parentheses are used. Probabilities are not functions, though they may be a function of different variables. Maybe instead of survival probability you sould just remove the 'S()' part and say that this is the linear predictor for the survival model? That would be more explicit about what that part of the table represents.

Table 3-5. defining delta AICc as delta AICc is not super helpful for people who may be a little unsure of this metric. Instead you could define it as the difference in AICc from the AICc score of the model with the relative best fit? You've also not said what the weight column is in these tables.

Table 6. You could separate the 95% CI's with a comma instead of |.

Figure 2. `ggplot2` defaults to grey axis text, which is not ideal. I don't use that package, but I think something like `theme_bw()` should update those so that they are black instead.

Figure 5. Please include 95% CI for these lines. Tick marks on the x and y axis would also be helpful.

6. PLOS authors have the option to publish the peer review history of their article (what does this mean?). If published, this will include your full peer review and any attached files.

Reviewer #1: No

Reviewer #2: **Yes: **Mason Fidino

---

## [Author Response · Author response to Decision Letter 0]

29 Sep 2023

Manuscript PONE-D-23-13153

Response to Reviewers

Dear Dr. Corti,

Thank you for giving us the opportunity to submit a revised draft of the manuscript “Using decoys and camera traps to estimate depredation rates and neonate survival” for publication in PLOS ONE. We appreciate the time and effort that you and the reviewers dedicated to providing feedback on our manuscript and are grateful for the insightful comments on and valuable improvements to our paper.

We have incorporated most of the suggestions made by the reviewers. Those changes are highlighted within the manuscript. Please see below, in blue, for a point-by-point response to the reviewers’ comments and concerns. All page numbers refer to the revised manuscript file with tracked changes.

Reviewers' Comments to the Authors:

Reviewer 1 indicated that you must conduct a proper test to compare with the other study conducted in the same area at the same time. Also, you should consider that predation rates may be affected by mother decisions that could be difficult to reproduce, then that must be discussed. 

Reviewer 2 has similar issues. This reviewer emphasizes that one problem is how the analysis was described in the methods section, but also that you should consider for uncertainty in predators’ presence and their activities.

Academic editor:

We note that Figure 2 in your submission contain copyrighted images. All PLOS content is published under the Creative Commons Attribution License (CC BY 4.0), which means that the manuscript, images, and Supporting Information files will be freely available online, and any third party is permitted to access, download, copy, distribute, and use these materials in any way, even commercially, with proper attribution. For more information, see our copyright guidelines: http://journals.plos.org/plosone/s/licenses-and-copyright.

Author Response: We were the ones who took the image used for Figure 1. We updated the image just in case with another one that we took . 

We note that Figure 1 in your submission contain [map/satellite] images which may be copyrighted. All PLOS content is published under the Creative Commons Attribution License (CC BY 4.0), which means that the manuscript, images, and Supporting Information files will be freely available online, and any third party is permitted to access, download, copy, distribute, and use these materials in any way, even commercially, with proper attribution. For these reasons, we cannot publish previously copyrighted maps or satellite images created using proprietary data, such as Google software (Google Maps, Street View, and Earth). For more information, see our copyright guidelines: http://journals.plos.org/plosone/s/licenses-and-copyright.

Author Response: The imagery in the photograph was either generated by one of our authors or is available to use with proper citation. We made sure to include the citation in the caption as well as the body in the text. 

Reviewer 1: 

1. This manuscript presents an interesting approach to quantify the percentage of ungulate neonate mortality determined by predation, based on experimental decoys and neonate odor. I have no technical issue to raise. 

Author response: Thank you.

2. However, I have two major comments. First, authors have found that potential estimate of mortality was comparable to those obtained through other studies based on the tracking of collared individuals. Although I am confident that the results of this comparison are reasonable, a proper test should be based on the comparison with a study conducted on the same population at the same time. This task would be clearly difficult to accomplish, but I think it should be acknowledged by authors in the Discussion also, describing the ecological conditions of the other studies used for comparisons. 

Author response: The studies we selected for comparison were within the same geographic region of the US as where we surveyed. Additionally, the study we used to compare costs and some survival estimates was conducted in a county that bordered our study area in North Carolina. As the reviewer noted, a true comparison would require running the decoy method concurrent with the capture track method within the same areas. We added a few sentences in the discussion to address this issue.

3. Second, it may be possible that natural predation rates be affected by ungulate selection of concealed sites for bedding/hiding neonates. The deployment of decoys by observers is assumed to mimic the selection of bedding/hiding sites by ungulate mothers or neonates which is hard to verify and which may influence comparisons. This issue should be considered and discussed thoroughly by authors.

Author response: Thank you for pointing this out. The reviewer is correct, and we added some sentences to describe this.

The revised text reads as “Maternal investment, such as direct protection and the selection of the fawn bedding sites, may influence the survival of fawns (28, 29). However, we still believe our approach is useful because depredation typically makes up most of the total neonate mortality (e.g., 88%, 14) and might be even higher given the risk of study related effects in capture-track studies. Although the decoy method cannot incorporate maternal investment, information on local maternal bedding site selection can help inform decoy placement in the most appropriate vegetation conditions. ” 

4. As a minor comment, Tables 4-5 should include standardized model weights.

Author response: Thank you for pointing this out. Based on reviewer comments we have redone the analysis with black bear range in the main models rather than separating. In doing this, we determined that black bear range was not a significant parameter and so removed table 4-5 as they are no longer relevant. 

Reviewer 2

In this paper the authors were interested in evaluating a new method to estimate neonate survival using camera traps The paper has a logical flow to it, which makes it easier to understand, though some of the paragraphs could use some restructuring to improve flow a little more (e.g., the introductions). 

Author response: We took your advice for the introduction section and restructured the first two paragraphs to have a better flow. (See Introduction section of response to reviewers for examples of change).

I though that the suggested method was interesting, and the authors could even go a bit further to sell the fact that this technique is non-invasive (brought up a few times at the start but really only referenced quickly again in the discussion).

Author response: We clarified the was non-invasive throughout the paper. Below are some updated lines that highlight that the decoy method is non-invasive.

However, I did have some concerns with the analysis. The biggest issue is how the analysis was described in the methods (see below), but also I wonder why the authors did not account for or at least explain why they did not account for uncertainty in predator presence on the landscape and also what the relative predator activity metric represents ecologically. 

Main point 1: The reporting of the methods leaves a lot of uncertainty with respect to how the models were fit, and what data was used. For Example, it is still unclear to me if there was a pre-decoy and post decoy model fitted to the data (i.e., models for each species) or if the authors included a decoy dummy variable. This is especially important to cover because the authors try to make inference about a decoy effect or rather that there is none. And I cannot determine how their model actually did this based on the covariates they said they included. This makes it so that a reader would not likely be able to recreate the authors analysis from the information presented, which is a problem. 

Author Response: Thanks for pointing out this deficiency, we have clarified our methods to be more specific on what we modeled and the difference between the pre-decoy and decoy sampling periods.

Main point 2: It is possible to account for uncertainty in predator presence on the landscape while fitting your survival models. There are a few ways you could do this. First, if you are interested in predator specific effects, you could just fit and occupancy models with a time to detection observation model. This is something covered in depth (model and code) by Kery and Royle (2015) and also something I used in Fidino et al. 2020, with data and code here). Conversely you could fit a standard occupancy model and then link that with the survival model. For a frequentist analysis, this could be done by using a bootstrap method 

Author response: The reviewer's suggestion is a clever way to look at more ecological effects of predator communities on fawn predation. While we appreciate this could provide additional ecological insights, we believe this recommendation is beyond the scope of this paper and would distract from our main objectives. Our focus was to: 1. test the functionality of using a decoy; 2. Determine if predators would interact with the decoy to obtain indices of predator-specific relative depredation rates; 3. test general survivorship information; and 4. Conduct a cost analysis. 

We believe that predator-specific effects would be a novel next step to test with the decoy method. We focused on the first predator interaction because after that initial or first few interactions, the decoy may have been damaged or dragged from the initial placement. Thus, bias would have been created if trying to determine predator specific related survivorship estimates. Additionally, we focused on first predator interactions regardless of species because it was more representative of a living fawn (i.e., a dead fawn could not be set up again for further interactions). We modeled sites with black bear and without differently due to strict management of black bear range in the state and because around half of survey locations were within the black bear range. 

Nonetheless, we agree this is an interesting possible extension of our approach and have added a few sentences to the discussion recommending this idea as a potential further investigation with the camera-decoy methodology. 

Reviewer 2 Line by line comments:

Abstract

1. Line 26: to split up the comas a bit it would be good to use em dashes for individuals less than 4 weeks old. 

Author response: We updated the commas to em dashses. 

2. Line 27: Population dynamics of a population is redundant. Just say “… can influence ungulate population dynamics” or something like that.

Author response: We revised to “and variation in their survival can influence ungulate population dynamics.”

3. Line 39: you can definitely report probabilities at percentages like this, but when you do that it gets a little weird when you need to report uncertainty (e.g. 95% confidence intervals). For example, 31% (95% CI – 22%, 35%) looks off given the multiple percentages. Instead, 0.31 (95% CI – 0.22, 0.35) looks much cleaner (and even saves space). 

Author response: We updated the abstract to include the confidence intervals for the survival probability and structured it to match the reviewer's recommendation.

Introduction 

1. The first paragraph could probably get split into two. The first half is about neonate survival in ungulates while the second half is about how researchers estimate neonate survival. As such, the topic sentence does not really line up with half of the content of this paragraph. You could get around this by adding a closing sentence after line 56 and then have the “consequently” sentence there be the topic sentence of a new second paragraph. Further, what is currently paragraph 2 in this draft has a mix of redundant information and more specificity than what is currently the second half of paragraph 1. If you split this paragraph, you can fold in a lot of info from the current second paragraph / remove redundancies.

Author response: We updated the first three paragraphs of the introduction to flow better and incorporate above suggestions.

Lines 54- 81.

2. Making predictions with the average model in this case is not good because the nest survival model does not use the identity link (at least in terms of calculating average parameters and then using those for predictions). This is covered at length by cade (2015). However, banner and higgs (2017) have some pointers on how this may actually be done. Chances are, if that authors used the standard approaches available, they first generated average parameter values and then used them for prediction. If that is the case, that is not the correct things to do with this model. ( I believe you can model average predictions instead. Furthermore, the notion that you can get an idea about the overall effect of a given parameter via model averaging is not really correct and this error is unfortunately quite pervasive in our field. You might instead consider that you have quite a few uninformative parameters in your competing models (e.g., if they are nested subsets of one another), and if that is the case, you may just need to work off a single model (see Arnold 2010). 

Author response: After looking at the suggested papers, we decided to remove the model averaging and make inferences based on the competing models, reoccurring parameters, and confidence intervals of those parameters. We also reran our models so black bear range was a binomial parameter rather than two unique models (black bear range or not black bear range). From this we found that black bear range was not even in the competing models so we removed the model results that separated sites based on black bear range. 

Introduction line by line comments

Line 52: … can have a greater influence… 

Author response: We made the change.

Lines 53-54: This sentence comes a bit out of left field. If variation in neonate survival is important, how does having reliable measurements of this variation inform adaptive management? Is it to control the variation? To increase survival? A little more clarity here would help to make sure the reader follows your logic. I don’t disagree with your statement, but right now you are forcing the reader to fill in some logical gaps in order to agree with this.

Author response: We updated the paragraph to provide more information in relation to the above statement.

Lines 54-59

Author response:

Line 56: you can drop the “also”

Author response: We made the change.

Line 67: Change ‘ An alternative method” to ‘Alternative methods’ as you list multiple methods here.

Author response: We made the change.

Line 72: Missing a closing sentence on this sentence.

Author response: We added “Alternatives for estimating the survival for hider species that are cost-effective, require less survey effort, and are non-invasive would benefit wildlife managers in obtaining neonate survival estimates.”

Line 82 – 83: … a cheaper and less intrusive method … right now there is uncertainty if “less” should be applied to the word ‘intrusive’

Author response: We changed less intrusive to non-invasive. 

Line 95-102: Since the topic here is predation it could also make sense to cite some papers that used decoy eggs for nest predations.

Author response: We added a relevant citation.

The source is Bravo C, Sarasa M, Bretagnolle V, Pays O. Detectability and predator strategy affect depredation rates: Implications for mitigating nest depredation in farmlands. Science of the Total Environment. 2022;829.

Line 104: Considering or assuming? You likely also need to defend this assumption / provide some logic behind it. Two come to mind to be :: as ‘hider’ species, if a predator gets within that distance they are 1) not likely to miss the prey and 2) the prey lacks a strategy to escape the predator at that distance.

Author response: We revised as recommended on lines 115-117.

Line 110: You’ve been past tense with this paragraph so it would read better to say ‘ we predicted that depredation rates…’ Also, why did you make this prediction?

Author response: We made the change.

Methods

Top – level thoughts

1. Why make predictions when more are present?

Author response: The other 3 predators (gray fox, domestic dog, and black vulture) are not main predators for fawns and are often excluded from other fawn-related mortality studies or just barely mentioned in the results section. We focused on the top 3 predators as they have been shown to dramatically influence white-tailed deer fawn survival across many studies. 

2. If the goal was to limit an individual predator visiting multiple camera trapping locations the clustered design only really achieves that at the “among-cluster’ scale for many of these species (i.e., 250 m is still quite close for larger ranging species like black bear or coyote). This may cause some small amount of non-independence among camera trapping locations, which at a minimum should be acknowledged here. 

Author response: We added a sentence to acknowledge that effect. Lines 149 – 151.

One thing that is not made clear in the methods is how the effect of neonate was quantified. I did not see any explanation about whether the first two weeks of data was used in the analysis or if that was simply done for acclimation. To me, it would seem like you could estimate weekly time to detection with all the data and include some binary variable to individuate a predation event (though there is some complication about what you could consider a predation even when the neonate decoy is not present). Regardless, some extra information here is welcome (not: looking at the results it seems like models were fit to each period separately, not seeing where this is brought up in the methods though I could have missed it)

Author response: We clarified that we used the initial 2-week survey period to determine a baseline relative count of potential predator information on lines 155 and 158.

Because there was no decoy and we had strict predator response categories, there would be no way to estimate weekly time to detection with all the data. Additionally, we only used the first predator event in survival models as some encounters left the decoy destroyed or carried off and we did not want to expose models to unnecessary uncertainty and bias. 

Using a single person a single person for camera trap surveys makes the assumption that this is safe to do so. 

Author response: Yes, each survey may require more than one technician. This particular study had one camera trapper thus reported as such. We clarified on lines 284-286.

Line by line comments:

Line 165: You say 1 dm here and than 10-cm everywhere else. Use one or the other

Author response: We made the change.

Line 184 – 190 : How did you measure 2.5 meters from the neonate in the videos?

Author response: We added clarification on lines 209-211.

Line 201 – 211: How was this measure (was not covered in the methods so far).

Author response: All covariate calculations were described in the above sections. How we calculated the vegetation-related sections was described in detail in the Vegetation survey sections. We added additional information to the camera section to highlight the differences between the pre-decoy and decoy surveys and updated the predator relative abundance calculation sentence. 

Line 232 – 233: Why fit the models separately for inside and outside black bear range? Coul dyou have included a dummy variable there for black bear (plus interactions with other covariates) instead? You’s have the same number of parameters to estimate, but you’d be sharing information for other species which could leave more precise estimates.

Author response: We reran models with all sites and a binomial covariate indicating black bear presence or not. Black bear range was not in the top ranked models thus we removed the tables investigating within and outside black bear range. We also updated the results and discussion section removing further discussion of separating the models between within and outside black bear range sites. 

Line 233 – 235 : So are we assuming that all possible combinations are biologically reasonable?

Author response: We selected these specific covariates (predator related and localized vegetation characteristics) as they are all ones that would influence whether or not a fawn would be killed naturally; thus we ran all possible combinations to get the better understanding if a certain subset of these covariates would highlight certain trends over others. 

Results:

1. What was the naïve occupancy of these different predators across the sample areas? Right now your model assumes that, aside from the black bear, these species are available to prey on your decoy (or more specially that they are present in each location). 

Author response: We did not model for occupancy. Our survival analysis was based on the first predator that interacted with the decoy and not on the individual species' effects on survival. Aside from black bear, all other predators were common throughout the whole survey area. Black bear range is heavily managed by the state wildlife agency, whereas the other species are not so we selected to investigate black bear range impact as half of the survey was in the range and half was not. 

Line by line comments:

Line 271 – 272: The reported detection rates are a little confusing because the second ones (0.0035 and 0.004) lack the / day component. 

Author response: We updated to include the /day on lines 281-282.

Line 272 -276: This was not covered in the methods, but did you fit the models separately ( a prey-decoy model and a post decoy model)? If you did, you did not actually make any statistical comparison about a decoy effect here, meaning this sentence is not correct. Second, if you did quantify this effect, failing to detect an effect does not mean that there is no effect (i.e., you should not say there was no change). Instead, you failed to detect a difference,. But again if these were separate models you did not quantify a difference. 

Author response: There is no model here. We were simply comparing the before and after detection rates (number of detections / total number of days) and the estimated error for species once the decoy was added. There is no way to test survival before the decoy was added as there would not be anything to sample because we did not have real fawns or some other basis to quantify survival. We updated the text to clarify that we cannot say we found no change. See lines 306-308 and lines 465 – 466.

Line 280 – 290: How were all these depredation probabilities calculated? Given the lack of uncertainty in the estimates, it seems like these are raw summaries of the data? However, if that is the case, these estimates would be biased as they do not account for the fact that your data was right-censored (which is what you nest survival model accounts for). These depredation probabilities can be made from your model, and as such could also come with an uncertainty estimate. For example, you could calculate the cumulative probability that a neonate gets predated over the course of your study for each species (or even the cumulative probability a neonate gets predated by any species, perhaps inside and outside of black bear range). 

Author Response: We calculated our relative depredation rates as # of predator specific kills / total number of deaths (Vreeland, Diefenbach, and Wallingford, 2004; Bravo et al. 2022). We updated the terminology to predator-specific relative depredation rates as our numbers are interested in the mortality information rather than including potentially “living” fawns. How we separated the calculation of depredation based on total kills and first kill was also done by Bravo et al. (2022) when using decoy eggs to test nest survival. We added more information to the methods section describing this information to attempt to clear up any confusion. See lines 214-222.

Discussion:

1. One thing that is not really brought up is that there is no doubt spatial variation in deer across the landscape. How are camera trapping sites selected relative to this? Are you only sampling in areas where deer are known to occupy with a high likelihood?

Author response: That is true that there is spatial variation in deer across the landscape. As many deer are game species, many locations have detailed information on their distribution and state and many county agencies have good information on local deer populations often from yearly hunting surveys so this information can be applied.

2. The concluding paragraph lists some future directions but does not bring us back to the contribution provided by this research. I’d circle back and remind the reader about key takeaways here and why this research helps move the field forward.

Author response: thank you for your suggestion. We added a sentence on line 480-482.

Line 369 – 370: How is the camera-decoy approach not perfect (i.e., give an example or two to the reader about this).

Author response: We removed this sentence, and the shortcomings of the decoy-camera method are later described in the discussion when describing areas of further needed research. 

Line 387 – 391: Again, I’m not sure that the authors specifically evaluated this effect given that they fit separate models to different subsets of the data, which means that they did not statistically evaluate / quantify a difference. Similarly, the authors calculated species – specific depredation rates from the raw data when it could have been estimated in the model. As such, the authors could make descriptive comparisons about their sample but it is a bit more difficult to assume that these values are representative of the larger population (i.e., using inferential statistics to get said estimates). As this paper is more methodological, some care should be taken here to not oversell what this technique can provide. For example, would it be necessary to get a much larger sample size to estimate predator specific effects? If so, what do the authors suggest?

Author response: We reran our models so it was one main model. We think testing predator effects in the survival models would be a great next way to test the decoy-camera methodology. Our depredation rates are based on predator specific relative “kills” over the total number of kills to test how well the decoy worked on each predator. We changed the terminology to predator-specific relative depredation rate and add a sentence describing directly how we calculated it into the method to try to clear up any confusion around the term. 

Tables and figures

1. All tables: Numeric columns should be right aligned and text columns should be left aligned. Right now it looks like the numeric columsn are center aligned.

Author response: We made the change.

2. Table 3-5 saying S= survival probability is a little odd. Looking at the models it appears to be a function in that parenthess are used. Probabilities are not functions, though they may be a function of different variables. Maybe instead of survival probability you should just remove the S() part and say that this is the linear predictor for the survival model? That would be more explicity about what that part of the table represents.

Author response: We made the change. 

3. Table 3-5. Defining deltaic as delta AiCc is not super helpful for people who may be a little unsure of this metric. Instead you could define it as the difference in AICc from the AICc score of the model with the relative best fit? 

Author response: We made the change.

---

## [Decision Letter · Decision Letter 1]

11 Oct 2023

Using decoys and camera traps to estimate depredation rates and neonate survival

PONE-D-23-13153R1

Dear Dr. Boone,

We’re pleased to inform you that your manuscript has been judged scientifically suitable for publication and will be formally accepted for publication once it meets all outstanding technical requirements.

Kind regards,

Paulo Corti, Ph.D.

Academic Editor

PLOS ONE

Reviewers' comments:

Reviewer's Responses to Questions

**Comments to the Author**

1. If the authors have adequately addressed your comments raised in a previous round of review and you feel that this manuscript is now acceptable for publication, you may indicate that here to bypass the “Comments to the Author” section, enter your conflict of interest statement in the “Confidential to Editor” section, and submit your "Accept" recommendation.

Reviewer #2: All comments have been addressed

2. Is the manuscript technically sound, and do the data support the conclusions?

Reviewer #2: Yes

3. Has the statistical analysis been performed appropriately and rigorously? 

Reviewer #2: Yes

4. Have the authors made all data underlying the findings in their manuscript fully available?

Reviewer #2: Yes

5. Is the manuscript presented in an intelligible fashion and written in standard English?

Reviewer #2: Yes

6. Review Comments to the Author

Reviewer #2: The authors did a great job addressing the comments I had on the previous draft of the manuscript. I've got no further comments. Good work on this!

- Mason Fidino

7. PLOS authors have the option to publish the peer review history of their article (what does this mean?). If published, this will include your full peer review and any attached files.

Reviewer #2: **Yes: **Mason Fidino

---

## [Editor Report · Acceptance letter]

16 Oct 2023

PONE-D-23-13153R1 

Using decoys and camera traps to estimate depredation rates and neonate survival 

Dear Dr. Boone:

I'm pleased to inform you that your manuscript has been deemed suitable for publication in PLOS ONE. Congratulations! Your manuscript is now with our production department. 

Kind regards, 

on behalf of

Dr. Paulo Corti 

Academic Editor

PLOS ONE